# Comprehensive Genome-Wide Analysis of the Receptor-like Protein Gene Family and Functional Analysis of *PeRLP8* Associated with Crown Rot Resistance in *Passiflora edulis*

**DOI:** 10.3390/plants13233264

**Published:** 2024-11-21

**Authors:** Weijun Yu, Fan Liang, Yue Li, Wenjie Jiang, Yongkang Li, Zitao Shen, Ting Fang, Lihui Zeng

**Affiliations:** 1Institute of Genetics and Breeding in Horticultural Plants, College of Horticulture, Fujian Agriculture and Forestry University, Fuzhou 350002, China; 15060537779@163.com (W.Y.); fanliang980430@126.com (F.L.); liyuehka@163.com (Y.L.); 15860691180@163.com (W.J.); leeykang97@gmail.com (Y.L.); ssszt08@163.com (Z.S.); fangting@fafu.edu.cn (T.F.); 2Fujian Academy of Forestry Sciences, Fuzhou 350012, China; 3Key Laboratory of Ministry of Education for Genetics, Breeding and Multiple Utilization of Crops, Fujian Agriculture and Forestry University, Fuzhou 350002, China

**Keywords:** passion fruit, receptor-like protein (RLP), crown rot, pathogen resistance

## Abstract

Passion fruit (*Passiflora edulis* Sims) is a Passifloraceae plant with high economic value. Crown rot caused by *Rhizoctonia solani* is a major fungal disease, which can seriously reduce the yield and quality of passion fruit. Receptor-like proteins (RLPs), which act as pathogen recognition receptors, are widely involved in plant immune responses and developmental processes. However, the role of *RLP* family members of passion fruit in resistance to crown rot remains unclear. In this study, evolutionary dynamics analysis and comprehensive genomic characterization of the *RLP* genes family were performed on passion fruit. A total of 141 *PeRLPs* in the genome of the ‘Zixiang’ cultivar and 79 *PesRLPs* in the genome of the ‘Tainong’ cultivar were identified, respectively. Evolutionary analysis showed that proximal and dispersed duplication events were the primary drivers of *RLP* family expansion. RNA-seq data and RT-qPCR analysis showed that *PeRLPs* were constitutively expressed in different tissues and induced by low temperature, JA, MeJA, and SA treatments. The *PeRLP8* gene was identified as the hub gene by RNA-seq analysis of passion fruit seedlings infected by *Rhizoctonia solani*. The expression levels of *PeRLP8* of the resistant variety *Passiflora maliformis* (LG) were significantly higher than those of the sensitive variety *Passiflora edulis* f. *flavicarpa* (HG). Transient overexpression of *PeRLP8* tobacco and passion fruit leaves enhanced the resistance to *Rhizoctonia solani*, resulting in reduced lesion areas by 52.06% and 54.17%, respectively. In addition, it can increase reactive oxygen species levels and upregulated expression of genes related to active oxygen biosynthesis and JA metabolism in passion fruit leaves. Our research provides new insights into the molecular mechanism and breeding strategy of passion fruit resistance to crown rot.

## 1. Introduction

To ward off threats from diverse pathogens, plants have evolved a two-layer immune system, including pattern-triggered immunity (PTI) and effector-triggered immunity (ETI) [1,2]. PTI is the initial layer of plant immunity and plays a key role when the pattern recognition receptors (PRRs) on the surface of plant cells detect pathogen- or microbe-related molecular patterns (PAMPs or MAMPs). PRRs are an important part of complex immune signal networks, which can sense various pathogens [3]. Recently, plant PRRs have been effectively employed to provide broad-spectrum resistance in pepper [4], poplar [5] and *Arabidopsis* [6].

The plant PTI response is primarily initiated by the recognition or perception of apoplastic elicitors by plasma membrane-localized pattern recognition receptors (PRRs). This recognition activates downstream immune responses, including the production of reactive oxygen species (ROS), extracellular Ca^2^⁺ influx, activation of mitogen-activated protein kinase (MAPK) cascades, and the expression of immune-related genes [7]. Based on structural characteristics, PRRs comprise receptor-like kinases (RLKs) and receptor-like proteins (RLPs), which lack a cytoplasmic kinase domain. In *Arabidopsis*, 57 RLP family members were receptor proteins located on the cell surface and composed of three distinct domains, including a leucine-rich repeat (LRR) domain on the extracellular surface, a transmembrane domain, and a short intracellular peptide instead of a kinase domain [8]. The first *RLP* gene discovered was *Cf-9* in tomato, which is involved in the immune response of tomato leaves against *Cladosporium fulvum* [9]. Further research has shown that members of the *RLP* family play a role in immune defense. For example, proteins encoded by the *Cf* and *Ve* genes in tomato, as well as the HcrVf2 protein in apple, were involved in mediating plant resistance to fungal pathogens [10]. Additionally, when the *TaRLP1* gene was knocked out, wheat exhibited reduced resistance to *Puccinia striiformis* f. sp. *Tritici* (*Pst*) and became more susceptible to infection. In contrast, overexpression of the *TaRLP1* gene enhances wheat’s resistance to *Pst*. These results suggested that *RLP* genes played a crucial role in plant response to pathogen infection.

Passion fruit (*Passiflora edulis* Sims) is a perennial evergreen vine-like fruit tree with high nutritional and economic value. However, passion fruit production is severely threatened by crown rot disease, which leads to significant yield losses and economic damage. When seedlings are infected with crown rot, dark brown lesions appear at the base of the stem, followed by gradual softening and rotting of the cortex, leading to brown rot development. Previous studies based on traditional morphological identification of fungal strains identified the primary pathogens responsible for stem rot in passion fruit as *Fusarium solani* and *Fusarium oxysporum*. In recent years, research on passion fruit has mainly focused on field symptoms, disease control measures, and cultivation techniques related to stem rot [11]. The chemical control methods were costly and harmful to the environment, making the development of disease-resistant cultivars one of the most effective strategies for managing passion fruit stem rot. However, there is limited in-depth research on the molecular mechanisms underlying crown rot resistance in passion fruit. In particular, the role of *RLP* genes in conferring resistance to crown rot is worth further studying.

In this study, *RLP* family genes were identified and characterized and a comparative evolutionary analysis was performed in the genome of the ‘Zixiang’ (ZX) and ‘Tainong’ (TN) cultivar. The orthologous and paralogous relationships were determined among these *RLP* genes. The chromosome distribution, gene structure, conserved motifs, and promoters of *PeRLP* genes were investigated. Moreover, a co-expression network with the *PeRLP* gene as the hub gene and positive response after being infected by *Rhizoctonia solani* was constructed. In addition, the *PeRLP8* gene was cloned from passion fruit, and its potential function was verified by RT-qPCR analysis and transient expression in wild tobacco and passion fruit leaves. Our study will provide potential candidate genes for the breeding of passion fruit resistant to crown rot through the genetic engineering method.

## 2. Results

### 2.1. Comprehensive Identification and Evolutionary Analysis of RLP Family Genes in Passion Fruit

Through excluding redundant sequences and genes encoding proteins with NB-ARC domains or kinase domains and verifying the *RLP* gene structures, 141 *RLP*-encoding genes in the *passiflra edulis* Sims (Zixiang, referred to as “ZX”) genome and 79 *RLP*-encoding genes in the *passiflra edulis* Sims (Tainong, referred to as “TN”) genome were identified. The number of two passion fruit *RLPs* had a significant expansion compared to the 57 *AtRLP* members in *Arabidopsis*. The *RLP* genes were renamed *PeRLP1*-*141* (ZX) and *PesRLP1*-*79* (TN) according to the location of the *RLP* genes to the respective chromosomes in the passion fruit genome. The 141 *PeRLPs* and 79 *PesRLPs* were distributed across nine chromosomes with varying numbers and uneven distribution (Appendix A). Many *PeRLP* genes were clustered in specific chromosomal regions in the ZX genome. For example, the highest number of *PeRLP* genes were found on Chr1 and Chr2, with 63 and 31 genes, respectively. Chr7 and Chr8 contain only 1 and 2 *PeRLP* genes, respectively. LG02 and LG05 had the highest number of *PesRLP* genes, with 13 and 31 genes in the TN genome, respectively. According to MCScanX results, dispersed and proximal duplications account for more than two-thirds of the duplication events in the ZX and TN genomes. These results indicated that the formation of these gene clusters and the expansion of *RLP* genes in passion fruit were likely due to these types of duplications.

To explore the evolutionary relationships among *RLPs* of eight plants, a phylogenetic analysis was conducted using Maximum Likelihood (ML) methods (Figure 1). The result showed that all *RLP* members cluster into nine distinct subgroups, labeled Group 1–9. This classification was supported by bootstrap values ranging from 60% to 100%, indicating the reliability of the *RLP* genes subfamily classification. A substantial number of *PeRLP* and *PesRLP* members were assigned to two groups (Group 5 and Group 8). Group 5 had the largest number of *RLP* members, including known disease resistance genes such as *HcrVf2*, *EIX2,* and homologous genes (35 *PeRLPs* and 21 *PesRLPs*) in passion fruit. Group 8 had the second largest number of *RLP* members, containing 50 *RLP* genes, but lacked homologs genes in *Arabidopsis* and tomato. Most *AtRLPs* were clustered together to form *Arabidopsis*-specific evolutionary branches (Group 3). These results suggested that *PeRLPs* and *PesRLPs* had undergone expansion following their divergence from a common ancestor with *Arabidopsis*.

### 2.2. Gene Structure Analysis of RLPs in Passion Fruit

Intron and exon structures play an important role in elucidating the phylogenetic relationships of gene families [12]. The exon-intron distribution pattern of *PeRLP* and *PesRLP* genes was further analyzed (Figure 2). The results showed that the exon number of *PeRLP* family genes ranged from 1 to 26, and the number of exons of *PesRLP* family genes ranged from 1 to 14. Most genes contain one or two exons. The exon number of *PeRLPs* and *PesRLPs* within the same group was also different in the evolutionary relationship. For example, *PeRLP1* had 12 exons, while *PeRLP119* had 9 exons in Group 1. These results provide a basis for the functional differentiation of the *PeRLP* and *PesRLP* gene families.

### 2.3. Conserved Domain Analysis of RLP Genes in Passion Fruit

To investigate the regulatory mechanisms of the *RLP* genes family in passion fruit, five highly conserved domains were identified (Figure 3). These domains were similar to those found in *RLP* genes of *Arabidopsis* and tomato [8,13]. Additionally, Multiple Em for Motif Elicitation (MEME) was used to further analyze the predicted RLP sequences to identify conserved motifs. A total of ten conserved motifs were identified among the predicted RLPs and were designated as motifs 1–10 based on e-values (Figure 3). The most common motif at the N-terminus was motif 10. Among 141 *PeRLPs* and 79 *PesRLPs*, 133 *RLP* genes were found to contain motif 10, accounting for 60.5% of the total number of *RLP* genes. No subgroup-specific motifs were identified among the nine subfamilies, indicating that these gene subfamilies may have originated from a common ancestor. Interestingly, the types and numbers of conserved motifs were relatively similar within each subfamily, indicating that there were potential functional similarities among genes within the same family.

Certain subfamilies exhibited similar motifs and motif sequences. For example, motifs 2, 3, 6, 7, and 8 were commonly found in the Group 9 subfamily, and motifs 7 and 10 were more prevalent across all subfamilies, indicating close phylogenetic relationships. Conversely, some subfamilies lacked specific motifs; for instance, *RLP* genes in Group 1 were missing motifs 3, 5, 6, 8, and 9. These results further supported functional differences among the Group 1–9 subfamilies. The difference in domain structures among *RLP* family members may contribute to diverse pathogen recognition and immune function, highlighting the multifaceted roles of passion fruit *RLPs* in pathogen response.

### 2.4. Collinearity Analysis of RLP Genes in Passion Fruit

A collinearity map was constructed to further explore the evolutionary trajectory of *RLP* members, and a list of collinear genes for *RLP* members across four species with chromosome-level genomes was generated (Figure 4). A number of highly conserved collinear blocks were shared among these genomes, with 74 of the 141 *PeRLP* genes (52.5%) mapped to these orthologous blocks. Clustering and identification of orthologous groups in ZX-TN and ZX-*Vitis vinifera* L. genomes revealed that their encoding genes exhibited similar intron distribution patterns and protein domain architectures. These results indicated that *RLP* orthologous groups among these different species may be functionally conserved. Additionally, 32 paralogous gene pairs in the ZX genome and 12 paralogous gene pairs in the TN genome were identified (Figure 4). Approximately 94% of these paralogous *RLP* genes displayed identical protein domain structures. Except for the *PeRLP70*-*PeRLP76* paralogous pair, all paralogous *RLP* gene pairs had *Ka*/*Ks* values of less than 0.9, indicating that most paralogous *RLP* genes had undergone purifying selection. These paralogous genes might be the extant product of whole-genome duplications (WGDs) during the evolutionary history of the Passifloraceae family.

### 2.5. Cis-Regulatory Element Assessment

*RLP* gene family members were involved in various developmental processes. To gain insight into their expression regulation, the 2000bp region upstream of each *PeRLPs* translation initiation site was analyzed using the PlantCARE database to identify potential transcription factor binding *cis*-elements. It was found that the promoter of *PeRLPs* contained *cis*-elements responsive to light, jasmonate (JA), methyl jasmonate (MeJA), abscisic acid (ABA), auxin, salicylic acid (SA), drought, defense, gibberellin, low temperature, circadian rhythms, and other factors (Appendix A). Among these, light-responsive *cis*-elements were the most common across all *PeRLPs* promoters. The *PeRLP126* promoter contained the highest number of *cis*-elements compared to other promoters. Notably, *PeRLP* promoters included *cis*-elements responsive to different plant hormones, especially had a higher number of MeJA (CGTCA and TGACG) and SA response elements, suggesting *PeRLPs* potentially respond to several hormones, such as SA, MeJA, and ABA. However, most *PeRLP* promoters contain single auxin (AuxREs, TGA-element, and TGA-box) and gibberellin response element. These findings indicated that *PeRLP* genes were more involved in stress responses, which were often primarily regulated by hormones, such as JA, SA, and ABA, rather than by growth hormones like auxin and gibberellin.

### 2.6. Expression Profile of PeRLP Genes

Gene expression profiles provide valuable resources for gene identification and functional analysis. The expression of 141 candidate *PeRLP* genes in flowers, leaves, roots, and seeds of passion fruit was analyzed by RNA-seq (Figure 5A). The TAU (τ) algorithm was used to determine the tissue-specific levels of 141 candidate *PeRLP* genes. It was found that 98 *PeRLP* genes had TAU values greater than 0.6, showing obvious tissue-biased expression patterns (Appendix A). Among them, 44 *PeRLP* family genes were preferentially expressed in passion fruit leaves, and 33 *PeRLP* family genes were preferentially expressed in the root (Appendix A). *PeRLP2*, *PeRLP104*, *PeRLP13*, *PeRLP107*, *PeRLP113*, *PeRLP135*, and *PeRLP139* were specifically expressed in the root (TAU value was 1), and the expression of *PeRLP134* in the root was much higher than that in other tissues. In addition, six *PeRLP* genes were only specifically expressed in leaves (TAU value was 1) (Appendix A).

The expression levels of all candidate *PeRLP* genes in the cold-sensitive variety ‘Huangjinguo’ (HJG) and cold-tolerant variety ‘Tainong’ (TN) under cold stress were calculated by reanalyzing the previously reported cold stress-free transcriptome data of passion fruit with reference to the ZX genome (Figure 5A). More than 10 *PeRLP* genes were found highly expressed under cold stress (Figure 5A). Five genes (*PeRLP83*, *PeRLP84*, *PeRLP85*, *PeRLP114*, and *PeRLP126*) did not have the expression. WGCNA was used to further identify the regulatory network of *PeRLP* genes involved in cold stress processes. A total of 16 co-expression modules were obtained. M7 and M14 modules were identified as highly correlated with cold tolerant of variety TN, suggesting the module members contained in them may be closely related to cold stress response (Figure 5B). A total of eight *PeRLP* hub genes belonging to the M7 module and one *PeRLP* hub gene belonging to the M14 module were identified by using WGCNA (Figure 5C). The co-expression network constructed with *PeRLP101* as the center included genes related to stress response, auxin signaling, and sucrose metabolism, indicating that *PeRLP101* may interact with these genes to actively respond to cold stress.

### 2.7. Expression Analysis of PeRLP Genes Under JA, MeJA, and SA Treatments

Given that most *PeRLP* family genes contain *cis*-acting elements related to hormone signaling, Six *PeRLP* gene promoter regions containing plant hormone response elements were selected to verify the transcriptional response to the treatments with JA, ABA, and MeJA hormones at 100 μmol/L concentrations (Appendix A). The results showed that six *PeRLP* family members differ in their specific responses to hormonal treatment. Notably, the expression levels of *PeRLP11* and *PeRLP70* changed most significantly under MeJA treatment, while *PeRLP8*, *PeRLP11*, and *PeRLP76* were significantly up-regulated within 0–8 h after JA treatment. The expression level of *PeRLP8* increased by nearly five times in 4 h compared with 0 h, indicating that it was more sensitive to JA treatment. In contrast, the transcription levels of *PeRLP17* and *PeRLP100* were significantly down-regulated under SA treatment. The expression patterns of *PeRLP8* and *PeRLP1* were opposite to those of *PeRLP70* and *PeRLP76* in response to SA treatment. The high sensitivity of these *PeRLP* genes under hormone treatments also validated the reliability of the predicted hormone-responsive *cis*-acting elements associated with them.

### 2.8. Expression Levels of PeRLP Genes in R. solani Infection

The crown rot of HG seedlings was identified as a disease caused by *R. solani*. The expression levels of all candidate *PeRLP* genes in the resistant cultivar *P*. *maliformis* (LG) *P. edulis* f. *flavicarpa* (HG) seedlings infected by *R. solani* were calculated using the ZX genome as a reference. Most of the *PeRLPs* belonging to Group 5 had high expression in the three stages of LG seedling infection with the bacteria (Figure 6A). WGCNA was used to further identify the regulatory network of *PeRLP* genes in response to the infection process of *R. solani*. For module-stage association, the MEblue module showed a high correlation with the L5 stage (Figure 6B). *PeRLP8* was identified as a hub gene in the MEblue module (Figure 6C). The co-expression networks of resistance-related genes include genes involved in ABA biosynthesis gene (*NCED2*), catabolism (*CYP707A*), and JA signal transduction pathway genes (*PP2C*), as well as other co-expressed genes involved in JA and ETH biosynthesis, signal transduction, sucrose metabolism and the key regulator of ROS (*RBOHD*). *PeRLP8* was also co-expressed with 16 *WRKY* family genes (Figure 6C). In addition, the expression pattern of PeRLP8 was supported by quantitative results (Appendix A). These results suggested that *PeRLP8* may interact with co-expressed genes to enhance LG’s resistance to *R. solani*.

### 2.9. Transient Expression of PeRLP8 Gene in Passion Fruit and Tobacco Leaves

To further study the role of *PeRLP8* in responding to *R. solani*, the CDS sequence of the *PeRLP8* gene was successfully cloned and sequenced. The constructed empty control 35S:: GFP and 35S:: *PeRLP8*-GFP were injected into passion fruit and tobacco leaves separately. Then, these leaves were inoculated with the fungal pathogen *R. solani*. It was observed that the passion fruit and tobacco disease leaves in the area of the leaves injected with 35S:: *PeRLP8*-GFP construct was significantly reduced by 54.17% and 52.06%, respectively (Figure 7A,B,D,E) and the expression of *PeRLP8* showed significantly up-regulated in injected areas compared to the control group (Figure 7C,F). These results suggested that the immune response was evoked after transient overexpression of the *PeRLP8* gene.

The physiological analysis showed that the content of ROS in passion fruit leaves overexpressing 35S:: *PeRLP8*-GFP was significantly higher than that of the control (Figure 7G). This suggested an enhanced oxidation reaction may contribute to the resistance to pathogen attack. In addition, the expression levels of three co-expressed genes related to the ROS signaling pathway (*PeRBOHD* gene) and the JA biosynthesis and signal transduction pathway (*PeMYC* and *PeJAR1*) were significantly up-regulated in passion fruit leaves with transient expression of *PeRLP8* (Figure 7H), verifying that *PeRLP8* may activate the defense responses via ROS and JA metabolic pathways.

## 3. Discussion

Plants sense pathogen infection through cell surface and intracellular receptors. *RLPs* are the main defense layer of the natural immune system against pathogen infection. The *RLP* gene family has been systematically studied in many plant species against pathogen infection [13,14,15]. However, *RLPs* in passion fruit have not been comprehensively analyzed. In this study, 141 *PeRLPs* and 79 *PesRLPs* genes were identified in the ZX and TN genomes, respectively. In terms of the number of genes, ZX and TN had significantly more genes than *Arabidopsis* which had 57 *RLP* genes [5,16]. This may be due to a WGD event in the Passiflora family [17,18]. Resistance gene families are often expanded through gene replication and diversification selection, which allows new pathogen recognition specificities to emerge [19]. In our study, the proximal and dispersed duplications were the main drivers of *RLP* family expansion in passion fruit. Furthermore, it has been reported that plants contain gene clusters of resistance gene analogs (RGAs), including *NLRs*, *RLKs,* and *RLPs* [13,20,21]. In tomatoes and peppers, several genes associated with RGAs were localized in gene clusters on different chromosomes [21,22,23]. Consistent with this, 141 *PeRLP* and 79 *PesRLP* members were found irregularly distributed across nine chromosomes and a large number of RLP genes formed gene clusters, which may be attributed to a single WGD event and proximal and dispersed gene duplication events. Additionally, the Group 4 branch of the *RLP* gene phylogenetic tree in different species contains several other *Cf* genes related to plant resistance to *C. fulvum* [24,25,26,27], Ve genes with race-specific resistance to tomato *Verticillium* [28], and 29 homologous *RLP* genes (*PeRLP8*, *PesRLP28*, *PesRLP79*, etc.) of passion fruit. This finding supported the view that the 29 homologous *RLP* genes that clustered with known *RLPs* involved in disease resistance may have a similar role in immunity.

Phylogenetic analysis showed that all *RLP* members in eight species clustered into nine different subgroups. Group 5 showed the major expansion and diversification of *PeRLPs* and *PesRLPs* from passion fruit, unlike *Arabidopsis*, which showed a sharp contraction of clade members with only one RLP. In contrast, all the genes of Group 3 belonged to *AtRLP* genes, which were greatly expanded and diversified. This phylogenetic analysis was consistent with the evolutionary analysis of *RLP* disease resistance genes in other plants [4,15,20]. The contraction or expansion of *RLP* genes in phylogenetic branches may be the result of plants coping with different selection pressures. Meanwhile, it is found that the conserved motifs of 141 *PeRLPs* and 79 *PesRLPs* members of ZX and TN were variable. It is reported that conserved motifs of genes were closely related to plants’ response to biotic and abiotic stresses [29]. These results suggested that passion fruit had formed its own immune receptor diversity under different selection pressures after being separated from its common ancestor.

Expression profiles were regarded as important clues for gene function analysis. *RLP* gene families in many plant species exhibit different expression profiles in different tissues or organs and regulate various biological processes [8,13,15,20]. For example, *RLP* genes specifically expressed in roots may play a role as a root-specific regulator of tomato [13]. There were 18 *MaRLP* genes expressed in the Cavendish cultivar roots [15]. This is consistent with our results that 6 *PeRLP* genes were specifically expressed only in passion fruit leaves (TAU value 1) and 33 *PeRLP* genes were expressed in passion fruit root. These results suggested that *PeRLP* genes were widely involved in biological functions during plant development, and may play an important role in coping with biological and abiotic stresses in passion fruit.

The occurrence of crown rot has become an important limiting factor for the development of the passion fruit industry. *RLP* genes were the main genes involved in plant defense response, which were involved in many aspects of life activities, including plant growth and development and resistance to exogenous pathogens [30,31]. It has been reported that the *AtRLP1* gene participated in the MAMPS pathway and could recognize bacterial eMAX factor [30]. The tomato *CF-9* gene was involved in the immune defense response of tomato leaves to Xanthomonas [9]. All the genes resistant to *Xanthomonas campestris* in tomatoes were cloned, and it was found that all the genes belonged to the *RLP* family [8]. These results indicated that *RLP* family members play a certain role in immune defense. In our study, WGCNA was used to analyze transcriptome data of passion fruit seedlings infected with *R. solani* and identified *PeRLP8* as a hub gene that was significantly up-regulated in pathogen-resistant variety LG compared to the sensitive variety HG during the infection of *R. solani*. Moreover, the leaf lesion area of tobacco and passion fruit with transient expression of 35S:: *PeRLP8*-GFP were reduced after infection with *R. solani* compared with the control. These results indicated that the *PeRLP8* gene played a role in the resistance of passion fruit to *R. solani*.

As a stress signal molecule, JA accumulated rapidly when plant tissues were invaded by pathogens or insects [32], and played an important role in the defense response to vegetative pathogens [33]. Previous studies reported that overexpression of SlRLP6/10 in tomatoes could induce the accumulation of ROS and increase the contents of JA and ET, thus enhancing tomato resistance to *P. infestans* [34]. In this study, the expression levels of *PeMYC* and *PeJAR1* which were co-expressed with *PeRLP8* were significantly up-regulated in passion fruit leaves when transformed with 35S:: *PeRLP8*-GFP, and the transcription level of *PeRLP8* was significantly up-regulated under exogenous JA treatment. These results indicated that *PeRLP8* may interact with the JA pathway and enhance the resistance to *R. solani* in passion fruit. In addition, it is reported that the expression of the *RBOH* gene can produce superoxide and other reactive oxygen species, which play a role in the host defense process [35]. ROS can induce plants to produce substances that inhibit the infection of pathogens and also induce plant defense response in signal pathway [36]. In our study, overexpression of *PeRLP8* led to significant up-regulation of the *PeRBOHD* gene, which was the key gene in the ROS biosynthesis pathway. Therefore, it was suggested that *PeRLP8* may promote the production of ROS in passion fruit leaves by regulating the *PeRBOHD* gene to improve the resistance of passion fruit to *R. solani*, which needs more experiments to be verified. Overall, these results suggested that the *PeRLP8* gene may played a role in the defense against pathogen through the synergies of ROS and JA metabolism in passion fruit.

## 4. Materials and Methods

### 4.1. Plant Materials

The passion fruit seedlings of crown rot resistant cultivar *Passiflora maliformis* (LG) and crown rot susceptible cultivar *P. edulis* f. *flavicarpa* (HG) were provided by the Genetic Breeding Laboratory of Horticultural Plant Genetics and Breeding, College of Horticulture, Fujian Agriculture and Forestry University, China, and forty-five LG and HG seedlings were used to be infected with *Rhizoctonia solani* (*R. solani*), respectively. The stem base samples of the two cultivars were collected at 1, 3, and 5 days after the seedlings were infected by *R. solani*. The samples collected at each stage after infection had three biological repeats, and each biological repeat consisted of five infected stem base samples. The material used for transient gene expression was derived from the leaves of wild-type tobacco and the ‘Qinguo No. 9’ variety passion fruit. According to previous studies, three exogenous hormones (JA, SA, and MeJA) with concentrations of 100 μmol/L were applied to six passion fruit seedlings of ‘Qinguo No. 9’, respectively [37,38,39]. The above samples were quickly frozen and stored in liquid nitrogen.

### 4.2. Identification of RLP Gene in Passion Fruit

“Zixiang” (ZX) and “Tainong” (TN) genome and available genome annotation data were downloaded from the National Genomics Data Center website (PRJCA004251 and PRJCA003078). Multiple strategies were used to search and identify members of the *RLPs* gene family, including keyword search, Hidden Markov Model search, and BLASTP search [4]. The BLASTP comparison was performed in the passion fruit genome by using 57 identified AtRLP sequences as reference. Next, the tBLASTn search was performed using the HMMER domain in the protein sequence of the passion fruit genome (threshold: 10^−4^). Combining the comparison results of HMM and BLASTP, a hypothetical *RLP* gene set was formed, which was manually screened to remove redundant sequences. The structure of *RLPs* was annotated using the Pfam database, and the genes with kinase and NB-ARC domains were filtered out with Pfam IDs (PF07714.12, PF00069.20, and PF00931). Finally, 141 *PeRLP* and 79 *PesRLP* genes were identified in ZX and TN genomes, respectively.

### 4.3. Distribution and Duplication of RLP Genes on Chromosomes in Passion Fruit

The distribution of *RLP* genes in the ZX and TN genome pseudochromosomes were mapped by the Advanced Circos program in the TBtools comprehensive toolkit [40,41]. The repetitive pattern of each *RLP* gene in the ZX and TN genomes was analyzed using BLAST search with E-value < 1 × 10^−5^. The BLAST results import MCScanX software (https://github.com/wyp1125/MCScanX (accessed on 15 August 2024)) for tandem and WGD duplication identification with the default parameter [42]. In addition, MCScanX was used to identify the collinear modules of *RLP* genes between species and species [42], and these replicators were classified by fragment replication, tandem replication, and random replication events within the species. The previous independent replicators were spliced into linear expansion pathways.

### 4.4. Phylogenetic Analysis of RLP Gene Family in Passion Fruit

MUSCLE was used to perform multiple sequencing on RLP protein sequences [12]. A phylogenetic tree of RLPs was constructed using IQ-TREE software (v2.0.3) with the maximum-likelihood (ML) method, JTT model, and a bootstrap value of 1000 [43].

### 4.5. Collinearity and Ka/Ks Calculation Analysis of RLP Genes

The One Step MCScanX-Super Fast plug-in in TBtools-II comprehensive toolkit was used to generate interspecies collinearity files for *Arabidopsis*, ZX, TN, and grape genome [40]. The coding sequence of tandem duplications for *PeRLP* gene pairs was selected to calculate non-synonymous (*Ka*) and synonymous (*Ks*) substitution ratios by the Neigojobori method [44].

### 4.6. Promoter Cis-Elements Analysis of RLP Genes

The PlantCARE was used to predict cis elements contained in the promoter sequence with the 2000 bp sequence upstream of the translation initiation codon of the *PeRLP* genes family [45], and the results were visualized via the iTOL website [46].

### 4.7. Gene Structure and Conserved Motif Analysis of RLP Genes

The coding sequence and UTR structure of *PeRLPs* from the original gff3 annotation file of ZX and TN genomes were visualized with TBtools-II [40]. The conserved motifs of the *PeRLP*s were predicted and analyzed through the MEME online website with default parameters [47].

### 4.8. The Expression Pattern of PeRLP Genes Family in Different Transcriptome Datasets

RNA-seq raw data of four passion fruit tissue samples (roots, leaves, seeds, and flowers) and cold-treated passion fruit seedlings were obtained from previous reports [17,48,49]. All of the above samples for RNA-seq had 3 biological replicates. The filtered data were mapped to the ZX genome by using the HISAT2 software (v2.10) with default parameters [50]. The StringTie software (v1.3.4) was used to standardize the expression levels of all genes in passion fruit [51]. Differentially expressed genes (DEGs) were screened and identified by the R package DESeq2 (https://github.com/thelovelab/DESeq2 (accessed on 15 August 2024)) and |log_2_foldchange| ≥ 1 and *p*-value < 0.05 parameters [52]. The tissue specificity index (TAC Calc) program in TBtools-II was used to calculate the tissue specificity of each *PeRLP* gene in 4 tissues with filter values greater than 0.6 [40]. WGCNA was used to identify resistance genes associated with cold tolerant variety TN [53]. According to our previous report [54], the hub *PeRLP* genes were screened with the correlation degree of module members greater than 0.9, and the co-expression network was constructed based on this. Coexpression networks were visualized by the Cytoscape software (3.7.1) [55].

*R. solani* was provided by the Horticultural Plant Genetics and Breeding Laboratory of Fujian Agriculture and Forestry University. In total, 45 seedlings of resistant variety LG and sensitive variety HG were, respectively, selected, and each seedling was infected with *R. solani*. With the stems of five seedlings as a biological duplication, three biological duplication stem samples were collected for RNA-seq. The WGCNA tool was used to identify the modules associated with the highly resistant variety LG for screening *PeRLP* genes in response to the infection of *R. solani*. Then, the hub *PeRLP* genes were screened with the correlation degree of module members greater than 0.9, and the co-expression network was constructed based on this. The reliability of RNA-seq was verified by quantitative detection of candidate *PeRLP* genes, according to the previous RT-qPCR description method [54]. Primers used in qPCR were listed in Appendix A.

### 4.9. Transient Expression Analysis of PeRLP8 in Tobacco and Passion Fruit Leaves

Amplification of *PeRLP8* CDS sequence was carried out using 2× Taq Master Mix (Vazyme, Nanjing, China). The resulting fusion products were inserted into a GFP-tagged pCAMBIA1302-GFP vector under CaMV35S promoter control by using Phanta Super-Fidelity DNA Polymerase (Vazyme, Nanjing, China) to form 35S:: *PeRLP8*-GFP recombinant vector. The heavy suspension of 35S:: *PeRLP8*-GFP recombinant plasmid was injected on the back of the leaves of tobacco and passion fruit with empty vector 35S:: GFP as the control. A total of 6 tobacco leaves (3 seedlings) and 6 passion fruit leaves (6 seedlings) were injected, respectively. The injected tobacco and passion fruit plants were cultured for 24 h in the dark environment, and cultured for 24 h under normal conditions. Then, the *R*. *solani* block with a diameter of 0.5 cm was inoculated on the leaves of tobacco and passion fruit expressing 35S:: GFP and 35S:: *PeRLP8*-GFP instantaneously, and the phenotype of the infected leaves was observed. The leaves of tobacco and passion fruit were collected on the third day after infection. There were 3 biological replicates with 2 leaves as a replicate. Tobacco and passion fruit leaves with transient expressions of 35S:: GFP and 35S:: *PeRLP8*-GFP were collected to measure expression levels of *PeRLP8*. In addition, the relative expression levels of three genes (*PeRBOHD*, *PeMYC*, and *PeJAR1*) co-expressed with the *PeRLP8* gene were also detected using qPCR in the transiently transformed passion fruit leaves. Primers used in qPCR were listed in Appendix A.

The contents of reactive oxygen species (ROS) were determined using ROS production rate test kit (COMIN, Suzhou, China) with spectrophotometry, as described previously [56].

### 4.10. Statistical Analyses

Statistical analyses were performed by Student’s *t* test or one-way ANOVA at a significance level of 0.05 using the SPSS 19.0. Figures were geminated by the GraphPad Prism 8.0.2 software.

## Figures and Tables

**Figure 1 plants-13-03264-f001:**
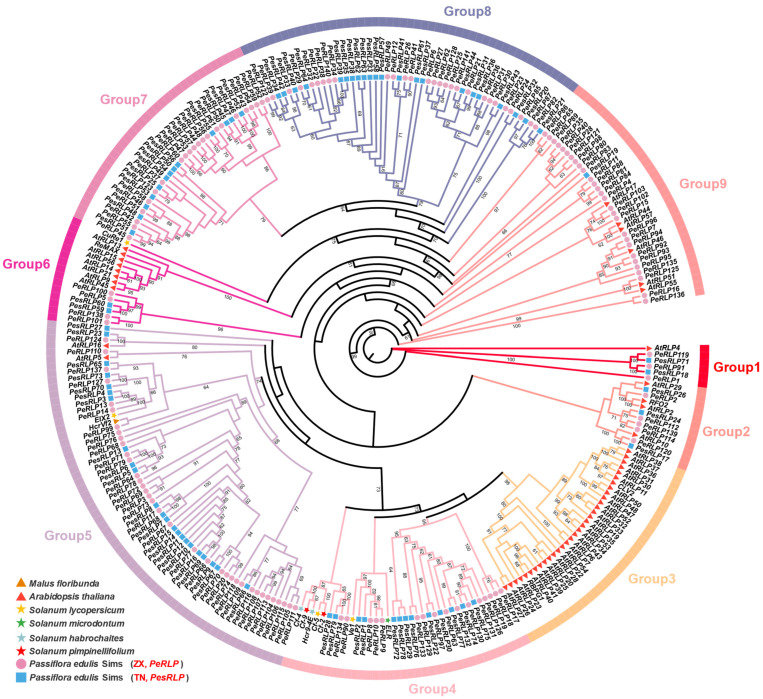
Identification and evolutionary analysis of *RLP* family genes in passion fruit. Phylogenetic analysis of the RLP homolog proteins from eight plant species, including *Malus floribunda*, *Arabidopsis thaliana*, *Solanum lycopersicum*, *Solanum microdontum*, *Solanum habrochaites*, *Solanum pimpinellifolium*, *passiflra edulis* Sims (ZX), and *passiflra edulis* Sims (TN). Each of the eight species was represented by a different shape in the evolutionary tree.

**Figure 2 plants-13-03264-f002:**
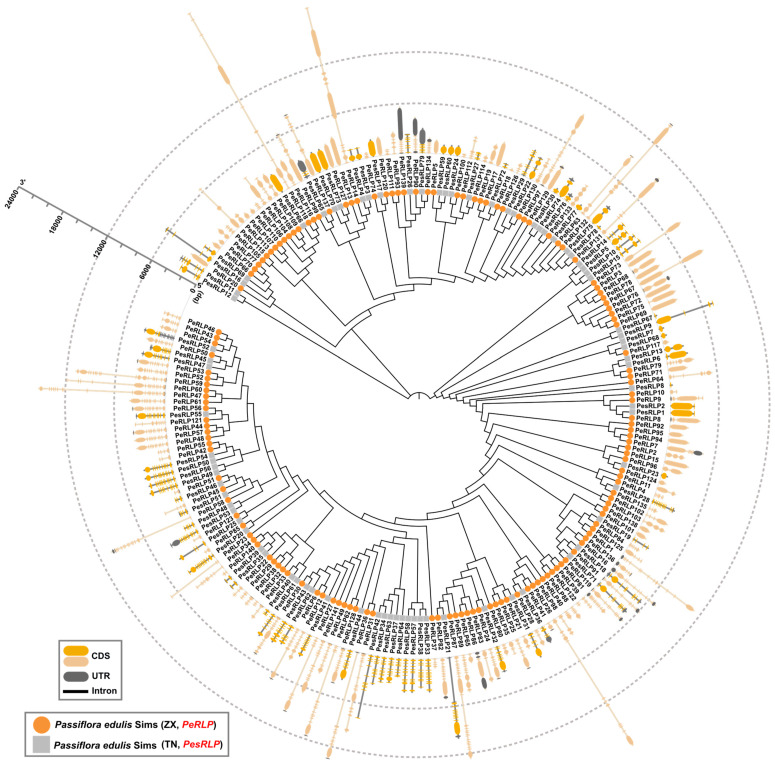
Gene structure of *RLP* genes in two passion fruit genomes. The circle represents 141 *PeRLP* genes in the ZX genome, and the square represents 79 *PesRLP* genes in the TN genome. The dotted line circles indicated the length of *RLP* genes.

**Figure 3 plants-13-03264-f003:**
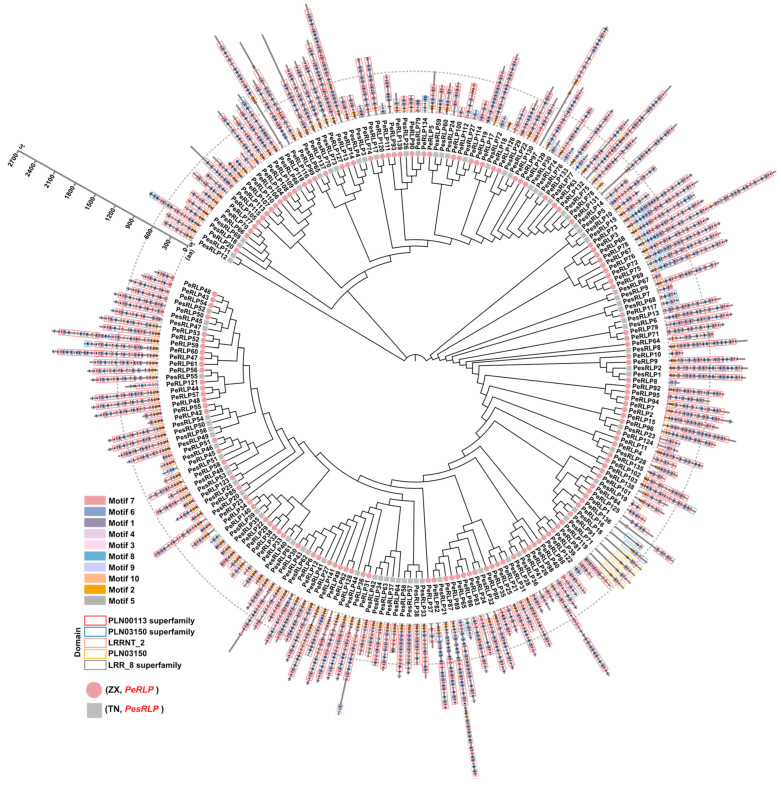
Conserved motif analysis of 141 PeRLP and 79 PesRLPs in the ZX and TN genomes. Different color modules indicated different motifs. Different color boxes indicate different conserved domains. The dotted line circles indicated the length of RLP proteins.

**Figure 4 plants-13-03264-f004:**
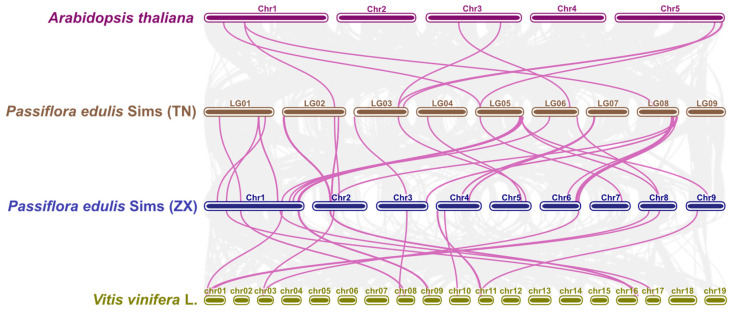
Synteny analysis of *RLP* genes in four plants, including *Arabidopsis*, *Passiflora edulis* Sims (ZX), *Passiflora edulis* Sims (TN), and *Vitis vinifera* L. genome. The purple lines display the collinear *RLP* genes among four plant genomes (*Arabidopsis*, TN, ZX and *Vitis vinifera* L.). The light gray lines represented collinear blocks.

**Figure 5 plants-13-03264-f005:**
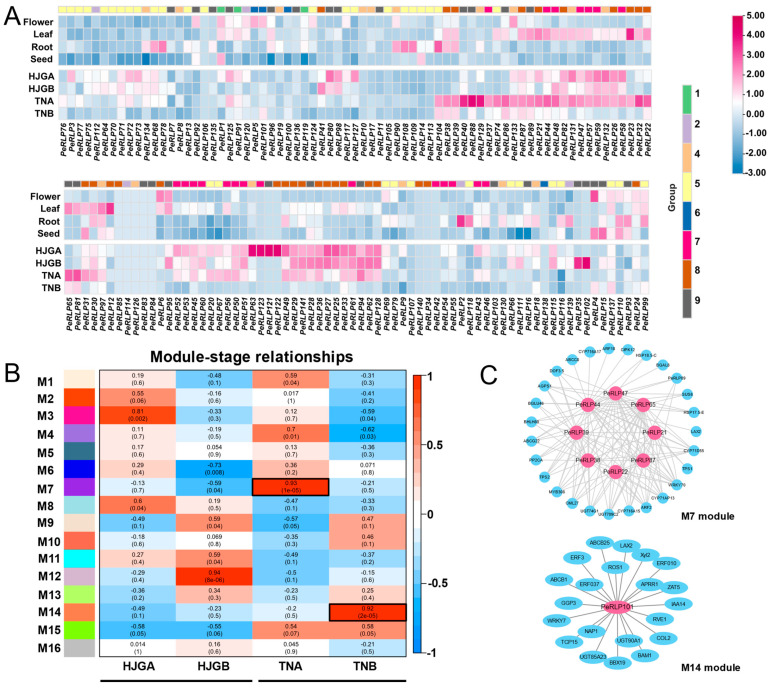
Expression profiles of 141 *PeRLP* genes. (**A**) Expression patterns of *PeRLP* genes in different tissues (root, leaf, seed, and flower) and seedlings of passion fruit with cold treatment. Three biological replicates of HJG (HJGA1, HJGA2, and HJGA3 were recorded as HJGA) and TN (TNA1, TNA2, and TNA3 were recorded as TNA) under normal temperature conditions. Three biological replicates of HJG (HJGB1, HJGB2, and HJGB3 were recorded as HJGB) and TN (TNB1, TNB2, and TNB3 were recorded as TNB) under cold stress. Differences in gene expression changes were shown in color as the scale, mediumvioletred for high expression, and steelblue for low expression. (**B**,**C**) Weighted gene co-expression network (WGCNA) was used to identify resistance genes associated with cold-tolerant variety TN. M1-M16 module indicated that the main branches constituted 16 merge modules (based on a threshold of 0.25) and were marked with different colors. Coexpression networks were constructed with eight and one *PeRLP* hub genes in the M7 and M14 modules, respectively.

**Figure 6 plants-13-03264-f006:**
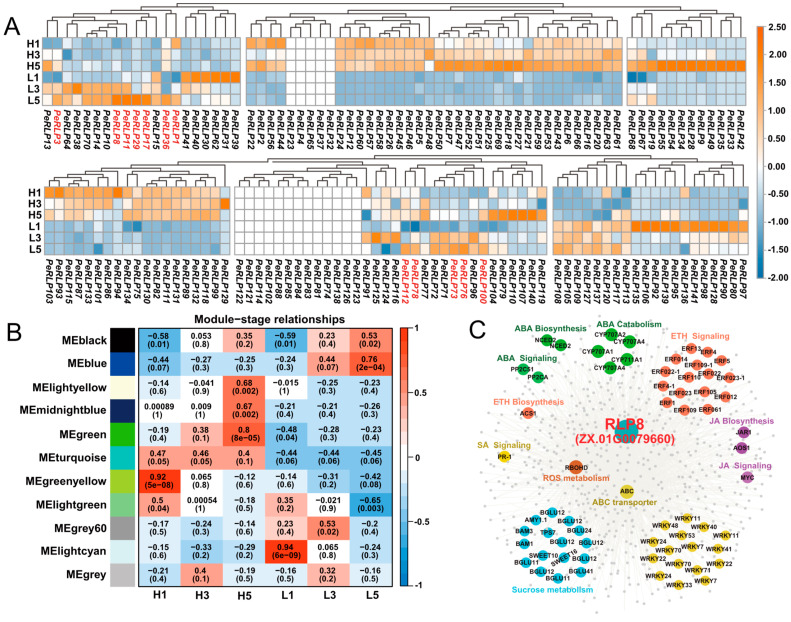
Analysis of gene co-expression network in RNA-seq data of passion fruit infected by *R. solani*. (**A**) Transcriptional expression analysis of 141 *PeRLP* during the infection of resistant variety LG and sensitive variety with *R. solani*. The heatmap was created based on the log2 (FPKM + 0.01) value of *PeRLP* genes and normalized by row. Differences in gene expression changes are shown in color as the scale, orange for high expression, and steelblue for low expression. L1, L3, and L5, respectively, represented the 1, 3, and 5 days after the resistant cultivar LG was infected by *R. solani*. H1, H3, and H5, respectively, represented the 1, 3, and 5 days after the sensitive cultivar HG was infected by *R. solani*. (**B**) The correlation analysis of modules and stages by using WGCNA, the heat map showed the correlation between modules and stages. Red and blue indicated positive and negative correlations, respectively. (**C**) The co-expression gene network was constructed with *PeRLP8* as the center in the MEblue module. SA signaling, Salicylic acid signaling; ETH biosynthesis and signaling, Ethene biosynthesis and signaling; ABA biosynthesis and catabolism, Abscisic acid biosynthesis and catabolism; JA biosynthesis and signaling, Jasmonic acid biosynthesis and signaling; ROS metabolism, reactive oxygen species metabolism.

**Figure 7 plants-13-03264-f007:**
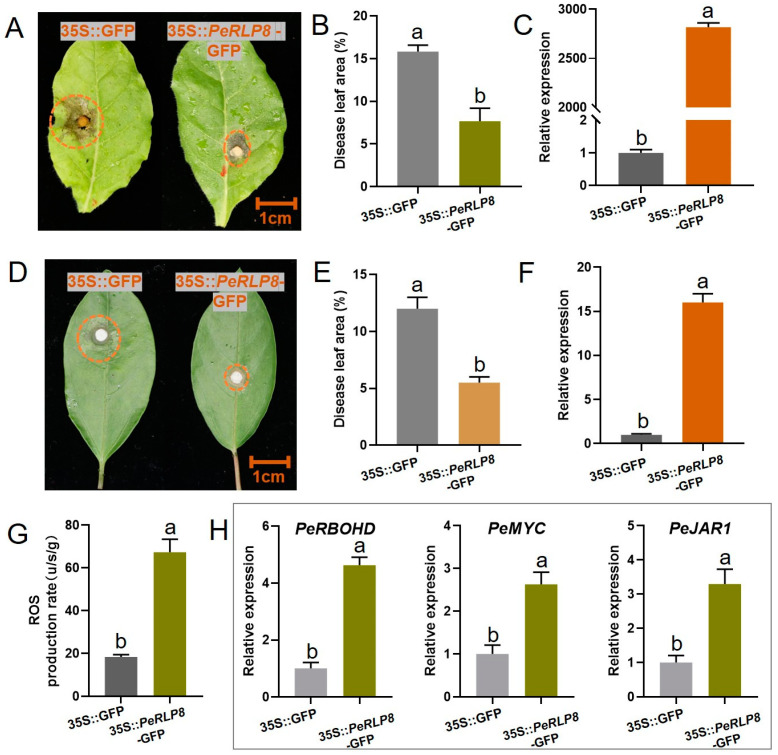
Resistance analysis of *PeRLP8*-overexpressed passion fruit and tobacco leaves against *R. solani.* The phenotype observation of transiently overexpressed passion fruit (**A**) and tobacco (**D**) leaves after inoculation with *R. solani*. The dotted lines on the leaves indicate the disease area. Bar = 1 cm. (**B**,**E**) The disease area statistics in *PeRLP8*-overexpressed passion fruit and tobacco leaves. (**C**,**F**) The expression levels of *PeRLP8* in transient-transformed tobacco and passion fruit leaves. (**G**) ROS content in transient-transformed passion fruit leaves. (**H**) Quantitative detection of 3 genes co-expressed with *PeRLP8* in *PeRLP8*-overexpressed passion fruit leaves. Different letters indicate statistically significant differences compared with the 35S:: GFP control (Student’s *t*-test, *p* < 0.05). The error bars represented standard error.

## Data Availability

The datasets generated during and/or analyzed during the current study are available from the corresponding author upon reasonable request.

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
