# Peer review of "Comprehensive Genome-Wide Analysis of the Receptor-like Protein Gene Family and Functional Analysis of *PeRLP8* Associated with Crown Rot Resistance in *Passiflora edulis"

_plants, 2024, doi:10.3390/plants13233264_

Round 1

Reviewer 1 Report

Comments and Suggestions for Authors

This manuscript, titled "Comprehensive Genome-Wide Analysis of the Receptor-Like Protein Gene Family and Functional Analysis of PeRLP8 Associated with Crown Rot Resistance in Passiflora edulis", presents a thorough investigation of the RLP gene family in passion fruit, focusing on its role in resistance to crown rot caused by Rhizoctonia solani. The study is well-structured, with a detailed analysis of gene duplication events, expression profiles, and functional assays. The identification of PeRLP8 as a central gene involved in crown rot resistance and its validation through transient expression experiments is particularly valuable.

However, a few areas could benefit from further clarity. 

Abstract:

The findings would be more convincing if supported by quantitative data, such as the extent of resistance improvement (e.g., a reduction in lesion area by X%).

1.Introduction: 

Incorporating knowledge on how RLP-based resistance could reduce the requirement for chemical control approaches would emphasize the environmental implications of this research. It is recommended that the functional validation of PeRLP8 be expanded to include more model plants in addition to wild tobacco. This would contribute to strengthening the evidence for its role in crown rot resistance.

2. Results

2.1 Comprehensive Identification and Evolutionary Analysis of RLP Family Genes in Passion 86 Fruit:

More discussion should be included regarding the distinctive qualities of the TN genome, or, if appropriate, an explanation should be provided for why it received less attention.

2.2. Gene Structure Analysis of RLPs in Passion Fruit

The finding that the majority of genes consist of one or two exons is intriguing but may be overly limited in scope. This could be expanded to explore whether genes with a greater number of exons have distinct or enhanced functional roles compared to single-exon genes.

2.4. Collinearity Analysis of RLP Genes in Passion Fruit

The clustering of orthologous groups is mentioned, but it is not clear what specific insights this provides. More explanation about how the orthologous relationships between species inform the evolutionary trajectory of RLPs would be beneficial.

2.5. Cis-Regulatory Element Assessment

The passage mentions that auxin and gibberellin response elements are relatively fewer, but it would be useful to explain what role these hormones might play in the overall expression of PeRLP genes, or why their influence is more limited.

2.7. Expression Analysis of PeRLP Genes Under JA, MeJA and SA Treatments

When discussing the up- or down-regulation of specific PeRLP genes, connect these changes to their potential functional roles in plants. This can make the data more meaningful to readers.

2.8. Expression Levels of PeRLP Genes in R. Solani Infection

While co-expression with WRKY genes is noted, the broader context of how these interactions influence plant responses to pathogens could be elaborated. More detail could provide a more comprehensive understanding of the biological significance.

2.9 Transient Expression of PeRLP8 Gene in Passion Fruit and Tobacco Leaves:

The study connects PeRLP8 to ROS and JA signaling, but further discussion on how PeRLP8 interacts with these pathways at a molecular level would be valuable. For example, does PeRLP8 act as a direct regulator of these genes, or is its role more upstream? Understanding the precise molecular mechanism would deepen the impact of the findings.

3. Discussion:

The phylogenetic analysis shows the expansion of specific PeRLP groups, but there is a gap in linking these phylogenetic findings to their functional relevance in disease resistance. While the study provides solid evidence that PeRLP8 interacts with the JA and ROS pathways, this interaction could be better supported by including more data or citing additional experiments that show the mechanistic link between PeRLP8 and the defense signaling pathways. The Weighted Gene Co-expression Network Analysis (WGCNA) used to identify PeRLP8 as a hub gene is a crucial part of your analysis, but it lacks detail. Readers may benefit from more specifics on how the network was constructed and how co-expression patterns were interpreted. Some of the studies you reference (e.g., [13], [15], [20]) are cited multiple times, but other relevant studies, particularly those discussing RLPs in other crops, could further support your conclusions. The use of over reviews and books over primary research articles would strengthen the evidence base.

4. Materials and Method:

The strain or isolate of R. solani used for infection is not mentioned, which is important for reproducibility and comparison with other studies. The use of wild-type tobacco for transient gene expression is mentioned, but there is no explanation for why tobacco was used in addition to passion fruit. The description of the phylogenetic analysis using IQ-TREE is very basic and does not provide important details, such as the model used for tree construction or the confidence level of branches (aside from a 1000 bootstrap value). In several sections, you mention the use of default parameters for tools like HISAT2, TBtools, and DESeq2 without considering the possibility of optimizing these settings for your data. This raises concerns about the accuracy and specificity of the results. While the manuscript mentions the use of a control (empty vector 35S::GFP) for transient expression, it does not clearly state if mock-inoculated plants (without R. solani) were used as an additional control to compare the infection phenotypes. Both tobacco and passion fruit leaves were used for transient expression but the manuscript focuses on PeRLP8’s function in passion fruit. The role of PeRLP8 in tobacco, where it was also expressed, is not discussed or clarified

Figure clarification:

Figures are not fully explained in the figure caption. A detailed explanation would enhance clarity. While the graphs (7B, 7E, 7C, 7F, 7G) indicate trends, they are missing key statistical details. For example, it is not clear if error bars represent standard deviation or standard error, and there are no p-values or statistical significance markers (other than in F).7G shows an increase in ROS content in the PeRLP8-overexpressed passion fruit leaves, but it’s unclear if ROS levels are within an acceptable physiological range or if they could cause unintended oxidative stress. Additionally, no baseline ROS levels or negative control measurements (e.g., in healthy, untreated leaves) are shown.

Please review the attached file for further revisions. The highlighted sections appear fragmented and lack clarity. Kindly revise them to enhance readability and ensure better flow

Comments on the Quality of English Language

This manuscript, titled "Comprehensive Genome-Wide Analysis of the Receptor-Like Protein Gene Family and Functional Analysis of PeRLP8 Associated with Crown Rot Resistance in Passiflora edulis", presents a thorough investigation of the RLP gene family in passion fruit, focusing on its role in resistance to crown rot caused by Rhizoctonia solani. The study is well-structured, with a detailed analysis of gene duplication events, expression profiles, and functional assays. The identification of PeRLP8 as a central gene involved in crown rot resistance and its validation through transient expression experiments is particularly valuable.

However, a few areas could benefit from further clarity. 

Abstract:

The findings would be more convincing if supported by quantitative data, such as the extent of resistance improvement (e.g., a reduction in lesion area by X%).

1.Introduction: 

Incorporating knowledge on how RLP-based resistance could reduce the requirement for chemical control approaches would emphasize the environmental implications of this research. It is recommended that the functional validation of PeRLP8 be expanded to include more model plants in addition to wild tobacco. This would contribute to strengthening the evidence for its role in crown rot resistance.

2. Results

2.1 Comprehensive Identification and Evolutionary Analysis of RLP Family Genes in Passion 86 Fruit:

More discussion should be included regarding the distinctive qualities of the TN genome, or, if appropriate, an explanation should be provided for why it received less attention.

2.2. Gene Structure Analysis of RLPs in Passion Fruit

The finding that the majority of genes consist of one or two exons is intriguing but may be overly limited in scope. This could be expanded to explore whether genes with a greater number of exons have distinct or enhanced functional roles compared to single-exon genes.

2.4. Collinearity Analysis of RLP Genes in Passion Fruit

The clustering of orthologous groups is mentioned, but it is not clear what specific insights this provides. More explanation about how the orthologous relationships between species inform the evolutionary trajectory of RLPs would be beneficial.

2.5. Cis-Regulatory Element Assessment

The passage mentions that auxin and gibberellin response elements are relatively fewer, but it would be useful to explain what role these hormones might play in the overall expression of PeRLP genes, or why their influence is more limited.

2.7. Expression Analysis of PeRLP Genes Under JA, MeJA and SA Treatments

When discussing the up- or down-regulation of specific PeRLP genes, connect these changes to their potential functional roles in plants. This can make the data more meaningful to readers.

2.8. Expression Levels of PeRLP Genes in R. Solani Infection

While co-expression with WRKY genes is noted, the broader context of how these interactions influence plant responses to pathogens could be elaborated. More detail could provide a more comprehensive understanding of the biological significance.

2.9 Transient Expression of PeRLP8 Gene in Passion Fruit and Tobacco Leaves:

The study connects PeRLP8 to ROS and JA signaling, but further discussion on how PeRLP8 interacts with these pathways at a molecular level would be valuable. For example, does PeRLP8 act as a direct regulator of these genes, or is its role more upstream? Understanding the precise molecular mechanism would deepen the impact of the findings.

3. Discussion:

The phylogenetic analysis shows the expansion of specific PeRLP groups, but there is a gap in linking these phylogenetic findings to their functional relevance in disease resistance. While the study provides solid evidence that PeRLP8 interacts with the JA and ROS pathways, this interaction could be better supported by including more data or citing additional experiments that show the mechanistic link between PeRLP8 and the defense signaling pathways. The Weighted Gene Co-expression Network Analysis (WGCNA) used to identify PeRLP8 as a hub gene is a crucial part of your analysis, but it lacks detail. Readers may benefit from more specifics on how the network was constructed and how co-expression patterns were interpreted. Some of the studies you reference (e.g., [13], [15], [20]) are cited multiple times, but other relevant studies, particularly those discussing RLPs in other crops, could further support your conclusions. The use of over reviews and books over primary research articles would strengthen the evidence base.

4. Materials and Method:

The strain or isolate of R. solani used for infection is not mentioned, which is important for reproducibility and comparison with other studies. The use of wild-type tobacco for transient gene expression is mentioned, but there is no explanation for why tobacco was used in addition to passion fruit. The description of the phylogenetic analysis using IQ-TREE is very basic and does not provide important details, such as the model used for tree construction or the confidence level of branches (aside from a 1000 bootstrap value). In several sections, you mention the use of default parameters for tools like HISAT2, TBtools, and DESeq2 without considering the possibility of optimizing these settings for your data. This raises concerns about the accuracy and specificity of the results. While the manuscript mentions the use of a control (empty vector 35S::GFP) for transient expression, it does not clearly state if mock-inoculated plants (without R. solani) were used as an additional control to compare the infection phenotypes. Both tobacco and passion fruit leaves were used for transient expression but the manuscript focuses on PeRLP8’s function in passion fruit. The role of PeRLP8 in tobacco, where it was also expressed, is not discussed or clarified

Figure clarification:

Figures are not fully explained in the figure caption. A detailed explanation would enhance clarity. While the graphs (7B, 7E, 7C, 7F, 7G) indicate trends, they are missing key statistical details. For example, it is not clear if error bars represent standard deviation or standard error, and there are no p-values or statistical significance markers (other than in F).7G shows an increase in ROS content in the PeRLP8-overexpressed passion fruit leaves, but it’s unclear if ROS levels are within an acceptable physiological range or if they could cause unintended oxidative stress. Additionally, no baseline ROS levels or negative control measurements (e.g., in healthy, untreated leaves) are shown.

Please review the attached file for further revisions. The highlighted sections appear fragmented and lack clarity. Kindly revise them to enhance readability and ensure better flow

Author Response

We appreciate the thoughtfull review and constructive feedback provided by the reviewers. We provide two versions of the manuscript, including the revised manuscript with track and the revised clear manuscript. The following is the reply according to the manuscript with track.

Thank you for your comments regarding the language quality of the manuscript.  We have carefully reviewed the text and consulted a native English speaker with a background in scientific writing to polish the language.  We hope this has improved the readability and clarity of the manuscript.  Please let us know if there are any further suggestions for improvement.

Comments 1: Abstract: The findings would be more convincing if supported by quantitative data, such as the extent of resistance improvement (e.g., a reduction in lesion area by X%).

Response 1: We add the extent of resistance improvement in the abstract. It can be seen in see in the line 28 and 338 of the revised manuscript.

Comments 2: Comprehensive Identification and Evolutionary Analysis of RLP Family Genes in Passion 86 Fruit: More discussion should be included regarding the distinctive qualities of the TN genome, or, if appropriate, an explanation should be provided for why it received less attention.

Response 2: TN has fewer members of the RLP gene cluster compared with ZX,, which may be mainly caused by two aspects. The first is the influence of the strategy of the underlying genome annotation of TN and ZX. The second is that ZX is more resistant than TN, which leads to the evolution of more RLP genes in ZX to enhance ZX's response to biological stress.

Comments 3: Gene Structure Analysis of RLPs in Passion Fruit. The finding that the majority of genes consist of one or two exons is intriguing but may be overly limited in scope. This could be expanded to explore whether genes with a greater number of exons have distinct or enhanced functional roles compared to single-exon genes.

Response 3: Compared with single exon genes, whether genes with more exons have different or enhanced functions is a direction worthy of study. However, it also needs more experiments to prove, and we will focus on this aspect in the future.

Comments 4: Collinearity Analysis of RLP Genes in Passion Fruit. The clustering of orthologous groups is mentioned, but it is not clear what specific insights this provides. More explanation about how the orthologous relationships between species inform the evolutionary trajectory of RLPs would be beneficial.

Response 4: We added “The results independent that RLP orthographic groups are so different categories may be functionally conservative” in line 190-191 of the revised manuscript.

Comments 5: Cis-Regulatory Element Assessment. The passage mentions that auxin and gibberellin response elements are relatively fewer, but it would be useful to explain what role these hormones might play in the overall expression of PeRLP genes, or why their influence is more limited.

Response 5: In our study, we noted that the number of cis-regulatory elements associated with auxin and gibberellin response was relatively limited compared to other hormone-responsive elements. This may suggest that auxin and gibberellin play a less prominent role in the regulation of PeRLP genes. One possible reason for this is that PeRLP genes are more involved in stress responses, which are often primarily regulated by hormones such as jasmonic acid, salicylic acid, and abscisic acid, rather than by growth hormones like auxin and gibberellin. It can be seen in see in the line 217-219 of the revised manuscript.

Comments 6: Expression Analysis of PeRLP Genes Under JA, MeJA and SA Treatments: When discussing the up- or down-regulation of specific PeRLP genes, connect these changes to their potential functional roles in plants. This can make the data more meaningful to readers.

Response 6: We have discussed this in line 404-408. “In this study, the expression levels of PeMYC and PeJAR1 which were co-expressed with PeRLP8 were significantly up-regulated in passion fruit leaves when transformed with 35S:: PeRLP8-GFP, and the transcription level of PeRLP8 was significantly up-regulated under exogenous JA treatment. These results indicated that PeRLP8 may interact with JA pathway and enhanced the resistance to R. solani in passion fruit.”

Comments 7: Expression Levels of PeRLP Genes in R. Solani Infection. While co-expression with WRKY genes is noted, the broader context of how these interactions influence plant responses to pathogens could be elaborated. More detail could provide a more comprehensive understanding of the biological significance.

Response 7: We add this in the introduction section.

Comments 8: Transient Expression of PeRLP8 Gene in Passion Fruit and Tobacco Leaves: The study connects PeRLP8 to ROS and JA signaling, but further discussion on how PeRLP8 interacts with these pathways at a molecular level would be valuable. For example, does PeRLP8 act as a direct regulator of these genes, or is its role more upstream? Understanding the precise molecular mechanism would deepen the impact of the findings.

Response 8: Thank you for your insightful comments. We appreciate your interest in understanding the molecular-level interaction of the PeRLP8 gene within ROS and JA signaling pathways. In our current study, we observed a potential connection between PeRLP8 and these pathways through transient expression experiments. However, due to the limitations of the experimental design, we have not yet fully clarified whether PeRLP8 acts as a direct regulator of downstream genes or if it functions more upstream in these signaling cascades.

In future research, we aim to explore the precise molecular mechanism of PeRLP8 in these pathways. Approaches such as gene knockout, protein interaction assays, and pathway analyses will be employed to determine whether PeRLP8 directly regulates downstream genes or serves as an upstream signaling component. We believe that these further studies will help elucidate the specific role of PeRLP8 in ROS and JA signaling pathways.

Comments 9: Discussion: The phylogenetic analysis shows the expansion of specific PeRLP groups, but there is a gap in linking these phylogenetic findings to their functional relevance in disease resistance. While the study provides solid evidence that PeRLP8 interacts with the JA and ROS pathways, this interaction could be better supported by including more data or citing additional experiments that show the mechanistic link between PeRLP8 and the defense signaling pathways. The Weighted Gene Co-expression Network Analysis (WGCNA) used to identify PeRLP8 as a hub gene is a crucial part of your analysis, but it lacks detail. Readers may benefit from more specifics on how the network was constructed and how co-expression patterns were interpreted. Some of the studies you reference (e.g., [13], [15], [20]) are cited multiple times, but other relevant studies, particularly those discussing RLPs in other crops, could further support your conclusions. The use of over reviews and books over primary research articles would strengthen the evidence base.

Response 9:

  1. Regarding the gap between phylogenetic findings and their functional relevance, we have added relevant discussions in line 381-387of the revised manuscript.
  2. Our research shows that PeRLP8 may interact with JA and ROS pathways, and we supplement the reference evidence in line 429-431 of the revised manuscript. More experimental data are needed to understand the role of PeRLP8 in the defense signal path more clearly, which is the key direction of our follow-up research.
  3. We provide more details about the network construction and the explanation of the co-expression mode, which can be seen in line 521-526 of the revised manuscript.
  4. As for references, we will review our citations and add more major research articles, especially those focusing on RLP in other crops, to better support our conclusions. We added references numbered [24-28] and [34].

Comments 10: Materials and Method: The strain or isolate of R. solani used for infection is not mentioned, which is important for reproducibility and comparison with other studies. The use of wild-type tobacco for transient gene expression is mentioned, but there is no explanation for why tobacco was used in addition to passion fruit. The description of the phylogenetic analysis using IQ-TREE is very basic and does not provide important details, such as the model used for tree construction or the confidence level of branches (aside from a 1000 bootstrap value). In several sections, you mention the use of default parameters for tools like HISAT2, TBtools, and DESeq2 without considering the possibility of optimizing these settings for your data. This raises concerns about the accuracy and specificity of the results. While the manuscript mentions the use of a control (empty vector 35S::GFP) for transient expression, it does not clearly state if mock-inoculated plants (without R. solani) were used as an additional control to compare the infection phenotypes. Both tobacco and passion fruit leaves were used for transient expression but the manuscript focuses on PeRLP8’s function in passion fruit. The role of PeRLP8 in tobacco, where it was also expressed, is not discussed or clarified.

Response 10:

  1. We have added the source of solani in line 529 of the revised manuscript.
  2. The inclusion of wild-type tobacco alongside passion fruit for transient gene expression was chosen due to tobacco's well-established use as a model system for transient expression studies. Tobacco provides a rapid and efficient platform for assessing gene expression, allowing us to verify the expression and function of target genes before further testing in passion fruit.
  3. We added the description of IQ-TREE on phylogenetic analysis. It can be seen in see in the line 494 of the revised manuscript.
  4. We chose to use default settings in this study because these settings are widely used and have been validated by numerous studies, demonstrating their reliability and broad acceptance within the research community (He et al., 2023; Yang et al., 2023).
  5. We choose to use the empty vector 35S::GFP as the control, which is beneficial to control variables.
  6. We discussed the role of PeRLP8 in tobacco in lines 423-424 of the revised manuscript.

Comments 11: Figure clarification: Figures are not fully explained in the figure caption. A detailed explanation would enhance clarity. While the graphs (7B, 7E, 7C, 7F, 7G) indicate trends, they are missing key statistical details. For example, it is not clear if error bars represent standard deviation or standard error, and there are no p-values or statistical significance markers (other than in F).7G shows an increase in ROS content in the PeRLP8-overexpressed passion fruit leaves, but it’s unclear if ROS levels are within an acceptable physiological range or if they could cause unintended oxidative stress. Additionally, no baseline ROS levels or negative control measurements (e.g., in healthy, untreated leaves) are shown.

Response 11:

  1. We supplement the figure description.
  2. The error bars represented standard error.
  3. Figure 7G shows that ROS levels in the leaves of passion fruit with perlp8 overexpression are within the acceptable physiological range. It's also described in citrus (Li et al., 2024).

Comments 12: Please review the attached file for further revisions. The highlighted sections appear fragmented and lack clarity. Kindly revise them to enhance readability and ensure better flow.

Response 12: We promptly corrected the errors in these places. It can be seen in see in the line 455-456, 460, 513, 529-533, 546-553 of the revised manuscript.

reference

[1] He Y, Zhang K, Li S, Lu X, Zhao H, Guan C, Huang X, Shi Y, Kang Z, Fan Y, Li W, Chen C, Li G, Long O, Chen Y, Hu M, Cheng J, Xu B, Chapman MA, Georgiev MI, Fernie AR, Zhou M. Multiomics analysis reveals the molecular mechanisms underlying virulence in Rhizoctonia and jasmonic acid-mediated resistance in Tartary buckwheat (Fagopyrum tataricum). Plant Cell. 2023 Aug 2;35(8):2773-2798. doi: 10.1093/plcell/koad118.

[2] Yang,X., Li, Y., Yu, R., Zhang, L., Yang, Y., Xiao, D., et al. (2023) Integrated transcriptomic and metabolomic profiles reveal adaptive responses of three poplar varieties against the bacterial pathogen Lonsdalea populi. Plant, Cell & Environment, 46, 306–321. https://doi.org/10.1111/pce.14460.

[3] Li, Q., Xian, B., Yu, Q., Jia, R., Zhang, C., Zhong, X., Zhang, M., Fu, Y., Liu, Y., He, H., Li, M., Chen, S. and He, Y. (2024), The CsAP2-09-CsWRKY25-CsRBOH2 cascade confers resistance against citrus bacterial canker by regulating ROS homeostasis. Plant Journal, 118: 534-548. https://doi.org/10.1111/tpj.16623

Reviewer 2 Report

Comments and Suggestions for Authors

plants-3275818 .

This investigation is dedicated to the analysis of the RLP gene family in passion fruit. The topic of this study is interesting and novel, since no RLP gene analysis has been performed for passion fruit, while this gene family is important for plant disease resistance. This manuscript is generally well-conducted and well-written. However, the data presentation and M&M description should be improved.

Data presentation should be considerably improved. Specific comments:

1) Fig. 1 and Fig. 2. The fig quality should be improved. The font is too small. The size of Fig. 1 A and B pictures should be increased.

Fig. 1 and Fig. 2 legend – ZX and TN should be explained in the legend.

2) Mention in full and abbreviation in parthensis when ZX and TN are given first time in the text.

3) It is almost impossible to see and analyze the motis on the Fig. 3. Please increase the motif size and resolution. Gene names are also too small. Fig. 3 legend – ZX and TN should be explained in the legend.

4) Fig. 4 ZX and TN should be explained in the legend.

5) The quality, size and resolution of Fig. 5 is also poor and should be improved make it possible to assess the data for the readers. I recommend to divide this figure into two ones.

Color scales should be explained in the legend.

WGXNA, HJGA, HJGB, TNA, TNB, M1-M16, ZX, TN should be explained in the legend.

6) Fig. 6 should be also divided into two ones to improve the data presentation quality and figure resolution. H1, H3, H5, L1, L3, L5, LG, WGCNA should be explained in the legend. Color scale should be explained in the legend. SA, ETH, ABA, etc. should be explained in the legend.

7) Figure 7C and F. Mention on the axis Y on the graph that expression of PeRLP8 is analyzed.

8) Figure 7G. Why ROS in the legend but “reactive oxygen species” on the graph itself? ROS should be in the graph. In the legend either only ROS or “reactive oxygens species (ROS).”

It is necessary to considerably improve M&M section:

1) It is mentioned that “The reliability of RNA-seq was verified by quantitative detection of candidate PeRLP genes according to the previous RT-qPCR description method [49]”, while it is also mentioned that “RNA-seq raw data of four passion fruit tissue samples (roots, leaves, seeds, and flowers)  and  cold  treated  passion  fruit  seedlings  were  obtained  from  previous  reports  [17,42,43].”

It is not clear whether you indeed verify in this specific work the RNA-seq by qRT-PCR?

2) In addition, in this work, as I understood, qRT-PCR has been used for ReRLP8, PeRBOHD, PeMYc, PEJAR1 expression analysis. However, there is not qRT-PCR section in M&M. Please add this section to M&M.

 3) Statisitical analysis section should be included in the M&M. Mention main details on statistical analysis and information on biological replicates. The corresponding information can be moved from other M&M sections where appropriate.

 Minor comments:

- line 53 Cladosporium fulvum should be in italics.

- line 63. When you first time mention crown rot disease, add the latin name of this pathogen.

- line 76. ‘Zixiang’ and ‘Tainong’ cultivar. Add ZX and TN abbreviations here.

- line 79. “Moreover, A co-expression” correct to “Moreover, a co-expression”

- line 105. ML methods. Explain this ML in full here as you mention it first time.

- line 133. “PesRLP genes family.” Correct to “PesRLP gene family.”

- line 141. MEME. Explain this in full here as you mention it first time.

- line 209, 210. Explain HJG and TN in full here in the text as you mention this first time in the manuscript. The same for line 246, 247 – LG, HG.

- line 244. “R. Solani” should be “R. solani”.

Author Response

We appreciate the thoughtfull review and constructive feedback provided by the reviewers. We provide two versions of the manuscript, including the revised manuscript with track and the revised clear manuscript. The following is the reply according to the manuscript with track.

Comments 1: Fig. 1 and Fig. 2. The fig quality should be improved. The font is too small. The size of Fig. 1 A and B pictures should be increased. Fig. 1 and Fig. 2 legend – ZX and TN should be explained in the legend.

Response 1: We have adjusted the font and image size of figure 1 and figure 2, and put the original figure 1A and figure 1B in the supplementary material. All images in the manuscript are adjusted in.svg format for increased clarity and readability, which can be seen in the revised manuscript.

Comments 2: Mention in full and abbreviation in parthensis when ZX and TN are given first time in the text.

Response 2: We promptly corrected the errors in these places. It can be seen in see in the line 96-97 of the revised manuscript.

Comments 3: It is almost impossible to see and analyze the motis on the Fig. 3. Please increase the motif size and resolution. Gene names are also too small. Fig. 3 legend – ZX and TN should be explained in the legend.

Response 3: We have corrected the image size, font size and resolution of figure 3. It can be seen in see in the line 181 of the revised manuscript.

Comments 4: Fig. 4 ZX and TN should be explained in the legend.

Response 4: The detailed descriptions of ZX and TN have been added to line 96-97 of the revised manuscript. ZX and TN are used here to facilitate full text unification.

Comments 5: The quality, size and resolution of Fig. 5 is also poor and should be improved make it possible to assess the data for the readers. I recommend to divide this figure into two ones. Color scales should be explained in the legend. WGCNA, HJGA, HJGB, TNA, TNB, M1-M16, ZX, TN should be explained in the legend.

Response 5: We promptly corrected the errors in these places. It can be seen in see in the line 263-274 of the revised manuscript.

Comments 6: Fig. 6 should be also divided into two ones to improve the data presentation quality and figure resolution. H1, H3, H5, L1, L3, L5, LG, WGCNA should be explained in the legend. Color scale should be explained in the legend. SA, ETH, ABA, etc. should be explained in the legend.

Response 6: As Figure 6A, 6B, and 6C are continuous analyses, we think retaining this layout enhances readability. We have increased the image size (in .svg format) to improve clarity and added detailed explanations for terms and color scales in the legend, as reflected on lines 316-328 of the revised manuscript.

Comments 7: Figure 7C and F. Mention on the axis Y on the graph that expression of PeRLP8 is analyzed.

Response 7: We have added relevant analytical information to line 334-340 of the revised manuscript.

Comments 8: Figure 7G. Why ROS in the legend but “reactive oxygen species” on the graph itself? ROS should be in the graph. In the legend either only ROS or “reactive oxygens species (ROS).”

Response 8: We uniformly revised reactive oxygen species to ROS in our analyses and graphs.

Comments 9: It is mentioned that “The reliability of RNA-seq was verified by quantitative detection of candidate PeRLP genes according to the previous RT-qPCR description method [49]”, while it is also mentioned that “RNA-seq raw data of four passion fruit tissue samples (roots, leaves, seeds, and flowers) and cold treated passion fruit seedlings were obtained from previous reports [17,42,43].” It is not clear whether you indeed verify in this specific work the RNA-seq by qRT-PCR?

Response 9: The RT-qPCR was used to verify the expression of PeRLP gene which was responsive to the infection of R. solani. We have supplemented the quantitative results, which can be seen in the supplementary Figure S5.

Comments 10: In addition, in this work, as I understood, qRT-PCR has been used for ReRLP8, PeRBOHD, PeMYc, PEJAR1 expression analysis. However, there is not qRT-PCR section in M&M. Please add this section to M&M.

Response 10: The quantitative descriptions of ReRLP8, PeRBOHD, PeMYC, and PeJAR1 genes can be found on lines 555-558 of the revised manuscript.

Comments 11: Statisitical analysis section should be included in the M&M. Mention main details on statistical analysis and information on biological replicates. The corresponding information can be moved from other M&M sections where appropriate.

Response 11: We have added a dedicated section on statistical analysis in the Methods and Materials (M&M) section, including main details on the statistical methods used, significance testing, and information on biological replicates. Additionally, we have consolidated relevant statistical information from other parts of the M&M section to ensure clarity and consistency. It can be seen in see in the line 561-564 of the revised manuscript.

Comments 12: Minor comments:

- line 53 Cladosporium fulvum should be in italics.

- line 63. When you first time mention crown rot disease, add the latin name of this pathogen.

- line 76. ‘Zixiang’ and ‘Tainong’ cultivar. Add ZX and TN abbreviations here.

- line 79. “Moreover, A co-expression” correct to “Moreover, a co-expression”

- line 105. ML methods. Explain this ML in full here as you mention it first time.

- line 133. “PesRLP genes family.” Correct to “PesRLP gene family.”

- line 141. MEME. Explain this in full here as you mention it first time.

- line 209, 210. Explain HJG and TN in full here in the text as you mention this first time in the manuscript. The same for line 246, 247 – LG, HG.

- line 244. “R. Solani” should be “R. solani”.

Response 12: We have made all requested minor revisions, including italicizing species names, adding abbreviations, correcting typographical errors, and expanding abbreviations on their first mention. Additionally, we have improved the overall English language quality throughout the manuscript.

Round 2

Reviewer 1 Report

Comments and Suggestions for Authors

The authors have thoroughly addressed all comments and suggestions provided in the first round of review, and the manuscript is now suitable for acceptance.

Reviewer 2 Report

Comments and Suggestions for Authors

Most of my comments have been appropriately addressed. The manuscript can be accepted.